# From Diagnosis to Treatment: Comprehensive Care by Reproductive Urologists in Assisted Reproductive Technology

**DOI:** 10.3390/medicina59101835

**Published:** 2023-10-15

**Authors:** Aris Kaltsas, Fotios Dimitriadis, Dimitrios Zachariou, Athanasios Zikopoulos, Evangelos N. Symeonidis, Eleftheria Markou, Dung Mai Ba Tien, Atsushi Takenaka, Nikolaos Sofikitis, Athanasios Zachariou

**Affiliations:** 1Department of Urology, Faculty of Medicine, School of Health Sciences, University of Ioannina, 45110 Ioannina, Greece; a.kaltsas@uoi.gr (A.K.); kzikop@uoi.gr (A.Z.); nsofikit@uoi.gr (N.S.); 2Department of Urology, Faculty of Medicine, School of Health Sciences, Aristotle University of Thessaloniki, 54124 Thessaloniki, Greece; difotios@auth.gr (F.D.); evansimeonidis@gmail.com (E.N.S.); 3Third Orthopaedic Department, National and Kapodestrian University of Athens, KAT General Hospital, 14561 Athens, Greece; dimitriszaxariou@yahoo.com; 4Department of Microbiology, University Hospital of Ioannina, 45500 Ioannina, Greece; eleftheria.markou4@gmail.com; 5Department of Andrology, Binh Dan Hospital, Ho Chi Minh City 70000, Vietnam; maibatiendung@yahoo.com; 6Division of Urology, Department of Surgery, School of Medicine, Faculty of Medicine, Tottori University, Yonago 683-8503, Japan; atake@med.tottori-u.ac.jp

**Keywords:** reproductive urologists, assisted reproductive technology (ART), male infertility, infertility evaluation, sperm retrieval, lifestyle factors, infertility treatment

## Abstract

Infertility is a global health concern, with male factors playing an especially large role. Unfortunately, however, the contributions made by reproductive urologists in managing male infertility under assisted reproductive technology (ART) often go undervalued. This narrative review highlights the important role played by reproductive urologists in diagnosing and treating male infertility as well as any barriers they face when providing services. This manuscript presents a comprehensive review of reproductive urologists’ role in managing male infertility, outlining their expertise in diagnosing and managing male infertility as well as reversible causes and performing surgical techniques such as sperm retrieval. This manuscript investigates the barriers limiting urologist involvement such as limited availability, awareness among healthcare professionals, and financial constraints. This study highlights a decrease in male fertility due to lifestyle factors like sedentary behavior, obesity, and substance abuse. It stresses the significance of conducting an evaluation process involving both male and female partners to identify any underlying factors contributing to infertility and to identify patients who do not require any interventions beyond ART. We conclude that engaging urologists more effectively in infertility management is key to optimizing fertility outcomes among couples undergoing assisted reproductive technology treatments and requires greater education among healthcare providers regarding the role urologists and lifestyle factors that could have an effect on male fertility.

## 1. Introduction

Infertility is a prevalent concern, impacting an estimated 10–15% of couples worldwide. Male factors contribute to approximately 20% of infertility cases. Moreover, they coexist with female factors in an additional 40% of cases [1]. Despite this, fewer men seek fertility solutions compared to women, and health professionals often overlook the need to refer males for further evaluation in a significant number of infertility cases. This discrepancy highlights the need for increased awareness and a more comprehensive engagement of urologists in the assessment and management of male infertility, especially in the current era of assisted reproductive technology (ART) [2].

The aim of this narrative review is to underscore the pivotal role of urologists in the evaluation and treatment of male infertility within the context of ART. The study also seeks to identify the barriers to the involvement of urologists in this process and suggests potential strategies to overcome these obstacles.

ART, which includes techniques such as intracytoplasmic sperm injection (ICSI), is increasingly utilized to tackle infertility [3]. The role of reproductive urologists in this process, however, is often overlooked. Reproductive urologists have the expertise to diagnose and manage male infertility, identify reversible causes, provide appropriate consultations, and perform surgical techniques such as sperm retrieval to enable ART or ICSI. Furthermore, they can recognize irreversible reasons for testis failure and provide guidance on the most suitable treatment options for couples seeking fertility assistance [4].

Despite the critical role urologists play in managing male infertility, several barriers limit their involvement. This study will explore these barriers in depth, including the limited availability and distribution of male reproductive urologists, particularly in certain regions. The lack of awareness and education among healthcare professionals and the general population about the role of urologists in male infertility contributes to the underutilization of their services. Additionally, financial constraints, including the absence of health insurance coverage for infertility diagnosis and treatment, also pose significant barriers to couples seeking care.

The role of urologists in the era of assisted reproductive technology is vital for diagnosing and managing male infertility. By highlighting the importance of urologists, exploring the challenges they face, and proposing potential solutions, this study aims to optimize the involvement of urologists in the management of male infertility and improve outcomes for couples seeking fertility treatment.

## 2. Materials and Methods

### 2.1. Study Design

This research was conducted as a narrative review, aiming to provide a comprehensive overview of the role of urologists in the management of male infertility within the context of assisted reproductive technology (ART).

### 2.2. Data Sources and Search Strategy

A systematic literature search was performed using the following electronic databases: PubMed, Scopus, and Web of Science. The search was conducted from the inception of each database up to April 2023 to capture relevant studies published within this timeframe.

### 2.3. Search Terms

The search strategy employed a combination of keywords and MeSH terms including “urologists”, “male infertility”, “Assisted Reproductive Technology”, “ART”, “sperm retrieval”, “diagnosis”, and “treatment”. Boolean operators (AND, OR) were used to refine the search.

### 2.4. Inclusion and Exclusion Criteria

Studies were included if they:

Were published in English;

Focused on the role of urologists in the diagnosis and treatment of male infertility;

Discussed the use of ART in the management of male infertility.

Studies were excluded if they:

Were not available in full text;

Were case reports, letters to the editor, conference abstracts, or reviews other than systematic reviews;

Did not provide relevant information on the role of urologists in the context of ART.

### 2.5. Study Selection and Data Extraction

Two independent reviewers screened the titles and abstracts of the identified studies. Full-text articles were then assessed for eligibility based on the inclusion and exclusion criteria. Any disagreements between reviewers were resolved through discussion or consultation with a third reviewer. Relevant data, including study design, sample size, main findings, and conclusions, were extracted from the included studies using a standardized data extraction form.

## 3. Understanding the Role of Urologists in Male Infertility Management amid the Advancements in Assisted Reproductive Technology

ART, encompassing techniques like in vitro fertilization (IVF), ICSI, and gamete intrafallopian transfer (GIFT), has revolutionized reproductive medicine by providing a path to parenthood for those struggling with infertility [5]. Infertility can result from both male and female factors. Yet, there is a widespread misconception that infertility is predominantly a female issue, overshadowing the importance of male infertility, which contributes to almost half of these cases [6].

The decreasing trend in male fertility, evidenced by the progressive reduction in sperm concentration over recent decades, underscores the need to focus on male reproductive health [7,8,9,10,11]. Lifestyle influences such as sedentary behavior, obesity, smoking, alcohol consumption, and substance abuse have been linked to impaired fertility in men. These factors can detrimentally affect sperm quality and motility, suggesting that adopting healthier habits can help improve fertility outcomes [12,13].

Reproductive urologists play an essential role in diagnosing and treating male infertility due to their specialized knowledge in male reproductive health. They are equipped to detect reversible causes of male infertility, such as varicoceles, ejaculatory duct obstruction, and hormonal imbalances, and to treat them effectively [14]. For instance, urologists can perform surgical procedures like varicocelectomy and testicular sperm extraction, which involves retrieving sperm directly from the testes [4]. In addition to offering these interventions, urologists can also provide advice on lifestyle modifications and prescribe appropriate medications to enhance fertility outcomes [15].

Despite advancements in the understanding and treatment of male infertility, several challenges hinder optimal patient care. These include limited access to specialized urological care, financial barriers, and a lack of awareness among healthcare providers and the public about the significant role of urologists in managing male infertility [16].

In conclusion, while ART has dramatically improved prospects for couples struggling with infertility, it is crucial not to overlook the significant contribution of male factors. Comprehensive care for infertility should include a strong focus on male reproductive health, lifestyle modifications that enhance fertility, and active involvement of urologists in infertility management. By addressing these areas, we can provide more effective care for couples aiming to achieve pregnancy [13].

## 4. The Urologist’s Role in Evaluation and Diagnosis

### 4.1. Evaluation and Diagnosis of Male Infertility

Reproductive urologists play a crucial role in the evaluation and diagnosis of infertility in couples undergoing ART treatments. The evaluation and diagnosis process involves a comprehensive assessment of both male and female partners to identify any underlying factors contributing to infertility. When the initial screening demonstrates abnormal semen parameters, abnormal male reproductive history, or couples with unexplained infertility, a complete diagnostic evaluation of the male partner should be performed by a urologist or male reproduction specialist [4,14]. This evaluation is important to improve a couple’s baseline natural fertility potential and to identify any male factor infertility issues that may require specific interventions or treatments [4,14].

The evaluation and diagnosis process typically includes a thorough medical history review, physical examination, and laboratory tests. The medical history review helps identify any potential risk factors or underlying medical conditions that may be contributing to infertility. The physical examination may involve assessing the external genitalia, checking for any abnormalities or signs of infection. Laboratory tests such as semen analysis, hormone testing, and genetic testing may be conducted to assess sperm quality, hormonal imbalances, or genetic abnormalities that could be affecting fertility [14].

In addition to the standard evaluation and diagnosis procedures, urologists may also utilize advanced diagnostic techniques to further assess male infertility. Testicular biopsy may be performed to evaluate testicular function and determine the presence of any abnormalities in sperm production [4]. Diagnostic testicular biopsy plays a pivotal role in the evaluation of male infertility, especially in distinguishing between obstructive azoospermia and non-obstructive azoospermia (NOA) and identifying intratubular germ cell neoplasia in situ (GCNIS) [14,17]. While there is a historical correlation between testicular biopsy histology and the likelihood of retrieving mature sperm cells [18,19,20,21], it is imperative to understand its clear indications grounded in evidence-based medicine [14,17].

The European Association of Urology (EAU) Sexual and Reproductive Health Guidelines advise against conducting testis biopsies, including fine needle aspiration, in isolation [14]. Instead, a simultaneous therapeutic sperm retrieval is recommended to eliminate the need for another invasive procedure post-biopsy [14]. The integration of diagnostic testicular biopsy with therapeutic testicular biopsy is essential for the cryopreservation process [14]. Notably, even individuals with severe spermatogenic failure, such as those with Sertoli cell-only syndrome (SCOS), might have localized areas where spermatogenesis is active [22,23].

Grounded in evidence-based medicine, it is imperative to understand the clear indications for performing a testicular biopsy. Regrettably, some medical professionals, notably from obstetrics/gynecology teams, have misapplied this intervention. Such misapplications are ethically and professionally indefensible, emphasizing the necessity for reproductive urologists equipped with specialized training to be the sole practitioners performing these procedures.

Genetic testing plays a crucial role in the evaluation of male infertility. For urologists specializing in reproductive medicine, a thorough understanding of prevalent genetic disorders associated with infertility is paramount to effectively counsel couples seeking fertility therapy. Men with azoospermia or low sperm counts may have opportunities for fatherhood through assisted reproductive technologies like IVF, ICSI, and testicular sperm extraction [14].

It is important to note that infertile males often exhibit a higher incidence of aneuploidy, structural chromosomal abnormalities, and DNA damage in their spermatozoa, which raises concerns about the potential transmission of genetic defects to offspring. The standard clinical approach involves analyzing genomic DNA derived from peripheral blood samples. However, assessing chromosomal abnormalities directly in sperm cells through sperm aneuploidy screening and preimplantation genetic testing can also be invaluable. Such tests are particularly recommended in specific scenarios, like cases of recurrent miscarriage [24,25,26,27,28,29,30].

Furthermore, urologists may collaborate with other specialists, such as reproductive endocrinologists, to ensure a comprehensive evaluation and diagnosis process. This collaboration is particularly important in cases where both male and female factors contribute to infertility. By working together, urologists and reproductive endocrinologists can develop a tailored treatment plan that addresses the specific needs of the couple and maximizes the chances of successful ART outcomes [4].

Overall, the role of urologists in the evaluation and diagnosis of infertility in the era of ART is crucial. They play a key role in identifying and addressing male factor infertility issues, which can significantly impact the success of ART treatments. Through a comprehensive evaluation process, urologists can provide valuable insights and recommendations for optimizing fertility outcomes in couples undergoing ART treatments.

Following the detailed procedures and evaluations, Figure 1 below provides a visual representation of the diagnostic work-up typically undertaken by a reproductive urologist during the evaluation and diagnosis of male infertility.

### 4.2. Identification of Patients Who Would Not Require Anything Other than ART

There are certain diseases for which no medical or surgical intervention can help. To save patients time and money, the reproductive urologist must first identify them [31].

#### 4.2.1. Oligoasthenoteratozoospermia

It can be difficult to determine when a male factor is a necessary indication for ICSI, although it is generally accepted that ICSI should be performed in men with severe male infertility. It is well known that conventional sperm testing is limited in assessing the severity of male infertility and recommending ICSI because it does not examine the functional components of sperm, including genomic integrity [32]. Whether ICSI is necessary instead of conventional IVF is often determined by measuring sperm count, motility, and morphology [33]. The live birth rates of ICSI and IVF in couples with oligoasthenoteratozoospermia have been compared, although high-quality data are lacking, as recently pointed out by Cissen and coworkers [34]. However, a meta-analysis of nine randomized control trials (RCTs) involving 332 treatment cycles in couples in which one partner had isolated teratozoospermia (two studies) or oligoasthenoteratozoospermia (seven studies) found that the risk ratio for achieving fertilization was 1.9 in favor of ICSI compared with conventional IVF (95% CI 1.4–2.5) [35]. The number of ICSI treatments required to prevent pregnancy failure in a single IVF cycle in this study was 3.1 (95% confidence interval (CI): 1.7–12.4) [35]. However, more recent data question these results, as no differences in fertilization, implantation, and pregnancy rates were found between the conventional IVF and ICSI groups in men with moderate oligoasthenoteratozoospermia (sperm concentration between 5 × 10^6^ per mL and 15 × 10^6^ per mL and progressive motility of 32%) [36].

#### 4.2.2. Isolated Teratozoospermia

It is common practice to evaluate sperm morphology data when selecting patients for ICSI. Kruger et al. were the first to propose using strict criteria to define sperm abnormalities and to recommend ICSI when the percentage of normal sperm in a given ejaculate was low [37]. The researchers found that the fertilization rate after traditional IVF was <8% in patients with abnormal sperm morphology (<4% normal) but >60% in patients with normal sperm morphology (4% to 14% normal) [38]. Later, other studies showed that for adequate fertilization rate in traditional IVF, at least 5% of the sperm must be morphologically normal [39,40]. Based on these results, ICSI has largely replaced traditional IVF as the treatment of choice for patients with teratozoospermia (defined as 5% morphologically normal sperm) [35,41]. However, a meta-analysis by Hotaling et al. examining the association between teratozoospermia and ART found no correlation between isolated teratozoospermia and IVF or ICSI success [42]. This meta-analysis included 2853 IVF/ICSI cycles involving 673 men with severe teratozoospermia (defined as less than 5% morphologically normal sperm) and 2183 men without this condition. Clinical pregnancy rates did not differ between couples in which the male partner had isolated teratozoospermia when either traditional IVF (OR 1.04, 95% CI 0.90–1.21) or ICSI (OR 0.95, 95% CI 0.63–1.42) was performed [42].

#### 4.2.3. Absolute Asthenozoospermia

ICSI is recommended for men with absolute (all sperm are immobile) or virtual (very few sperm are motile) asthenozoospermia [43]. Ultrastructural abnormalities of the sperm tail or complete necrozoospermia (no live sperm in the ejaculate) are the most common causes of the first form, whereas freezing damage sometimes occurs in the second form. Assessment of sperm viability is critical in extreme asthenozoospermia because injection of uncharacterized, immobile sperm is associated with lower fertilization and pregnancy rates [43]. Exposure of sperm to pentoxifylline or theophylline has been suggested to increase motility, while hypoosmotic threshold testing and the use of lasers have been recommended to increase viable sperm selection [44]. If none of these strategies are effective, sperm can be collected from the testes and then used in ICSI techniques [45].

#### 4.2.4. Antisperm Antibodies

Antisperm antibodies (ASAs) often develop in semen as a result of a breach in the blood–testicular barrier or a blockage in the male reproductive system. According to one study, three to twelve percent of men undergoing evaluation for infertility had abnormally high levels of ASAs in their semen [46]. Testicular torsion, testicular surgery, vasectomies, epididymo-orchitis, testicular cancer, cryptorchidism, and human immunodeficiency virus (HIV) infection have been associated with this disorder. The effects of ASAs on sperm motility, sperm capacitation, acrosome response, and sperm–oocyte binding have been associated with decreased fertility [46]. Antibodies directed against sperm can also inhibit sperm functionality by triggering cytokine production [47]. In couples undergoing conventional IVF or ICSI, a 2011 meta-analysis that included data from 16 observational studies totaling 4209 ART treatments (1508 IVF and 2701 ICSI cycles) analyzed levels of ASAs. Pregnancy rates were similar between traditional IVF and ICSI in men with high ASA values (odds ratios for failure to conceive were 1.22 (95% CI 0.84–1.77) and 1.00 (95% CI 0.72–1.38), respectively) [48].

#### 4.2.5. Globozoospermia

Globozoospermia is very rare, affecting only about 0.1% of andrologic patients. It is characterized by the absence of the acrosome of the sperm, which gives them their characteristic round shape. Depending on how many sperm have a spherical head, it is referred to as a total or partial form. Phenotypic criteria may not be used for diagnosis. To date, there have been few reports of paternity in globozoospermia, and none of them have been confirmed by deoxyribonucleic acid (DNA) testing. There is general agreement that ICSI is the gold standard therapy for globozoospermia [49].

#### 4.2.6. Female Aging

Women’s fertility declines, especially beyond the age of 30. This decline begins around the age of 30, continues until the age of 35, and accelerates after the age of 35. Over time, we naturally lose some of our ability to conceive, and there is currently no way to reverse this [50]. Anti-mullerian hormone (AMH) is an excellent indicator of ovarian reserve and a predictor of successful ART outcomes [51]. Any therapy for male infertility would be useless if the woman is over 40 years of age or has a low antral follicle count (AFC).

### 4.3. Identifications of Situations Where Donor Insemination or Adoption Is the Only Solution

#### 4.3.1. Y Microdeletions

Most cases of severe oligozoospermia and azoospermia can often be traced back to a molecular genetic defect. This defect is typically caused by deletions in clinically significant regions of the azoospermia factor (AZF) gene [52]. These Y-linked microdeletions are classified into three distinct categories: AZFa, AZFb, and AZFc [53].

No Yq microdeletions have been detected in normozoospermic men, so a causal relationship between Y deletions and infertility has been established [54]. Y deletions are more common in men with azoospermia (8–12%) than in men with oligozoospermia (3–7%) [55,56]. Sertoli cell-only syndrome is the result of loss of the entire AZFa region, while spermatogenic arrest is the result of loss of the entire AZFb region. In patients with complete deletions in the AZFa and AZFb regions, sperm retrieval is unsuccessful at the time of testicular sperm extraction (TESE). Therefore, these patients are not good candidates for TESE, and the available fertility options are the use of donor sperm or adoption [57,58]. Deletion in the AZFc region can range from azoospermia to oligozoospermia. Men with AZFc microdeletions have sperm in 50–75% of cases [57,58,59].

#### 4.3.2. XX Male Syndrome

When assessing male fertility in otherwise healthy men, urologists rarely encounter the 46,XX disorder of sex development (DSD), a genetic abnormality. Such 46,XX maleness is a rare disease that affects only 1 in 20,000 men and is characterized by a greater or lesser discordance between phenotype and genotype [60]. The development of testes in people with two X chromosomes and no normal Y chromosome is indicative of XX masculinity. All of these individuals are infertile, although 90% of them have typically male external genitalia and 10% have undescended testis, micropenis, and hypospadias [61]. In more than 80% of cases, the male phenotype persists because of a translocation of the sex-determining region Y (SRY) gene to a sex chromosome or autosome [62,63,64,65]. The remaining patients who test negative for SRY may have inhibitors of the male pattern mutation or mosaicism for the SRY gene [62,63,64,65]. Although SRY-positive individuals rarely present with genital abnormalities, they often show the phenotypic features of Klinefelter syndrome, such as hypogonadism, gynecomastia, azoospermia, and hyalinization of the seminiferous tubules, all of which are associated with altered hormone levels during puberty (low testosterone, elevated follicle-stimulating hormone (FSH), and luteinizing hormone [LH]) [66].

Physical examination may reveal some telltale signs, but these are not sufficient for a definitive diagnosis. Patients with 46,XX DSD often have short stature because of the absence of the testosterone-related pubertal growth spurt or the absence of additional Y-chromosome-specific growth hormones, in contrast to Klinefelter patients (47,XXY), who may have comparable problems [67]. Physical examination is the only means to diagnose testicular atrophy and other less common conditions such as undescended testis and hypospadias [68]. Azoospermia is present on sperm analysis even if all Y-chromosome AZFs are absent [53]. Hypergonadotropic hypogonadism is detected by serum hormone testing, which indicates primary testicular insufficiency. If a Y chromosome cannot be identified in a phenotypically male individual by karyotype analysis, fluorescence in situ hybridization (FISH) or molecular amplification by polymerase chain reaction (PCR) can be used to test for the presence or absence of the SRY gene. This evaluation does not predict prognosis but provides information on the numerous genomic rearrangements associated with the disease [69].

Once the diagnosis is made, it is best to have the patient treated by a team of specialists. The patient and their partner will need genetic counseling to better understand their situation and may also benefit from psychological counseling to better process what may be difficult information. Testicular biopsy for possible intracytoplasmic sperm injection is not helpful in individuals with typical infertility because they lack germ cell components. Artificial insemination, in vitro fertilization with donor sperm, or adoption are the only options related to fertility [70].

#### 4.3.3. Testis-Expressed 11 Gene

Testis-expressed 11 (TEX11), a member of the testis-expressed gene family, plays a crucial role in the formation of crossovers and the synapsis of chromosomes during the process of meiosis [71]. Overall, TEX11 mutations have been found in up to 15% of individuals with meiosis arrest and in more than 1% of azoospermic males. Reproductive disorders caused by meiosis arrest in NOA males are due to recessive mutations in this gene [71,72,73]. Recent studies by Krausz and colleagues have shown that abnormalities of the gene in humans result in complete metaphase arrest, as evidenced by persistent spermatocyte development and a sharp increase in the number of apoptotic metaphases [74]. It is virtually impossible for a man with a loss-of-function TEX11 mutation to have mature sperm in his testes. The only options for the couple are adoption or sperm donation if they want to have children [75].

## 5. The Pivotal Role of the Reproductive Urologist in Addressing Male Infertility and Enhancing ART Outcomes

Reproductive urologists play a pivotal role in diagnosing and managing male infertility. They skillfully identify and treat causes of infertility that are reversible, assess irreversible conditions that can be managed with ART, and provide a range of solutions from surgical procedures to medication (and even lifestyle adjustments), all aimed at optimizing fertility outcomes. To encapsulate these multifaceted strategies, please refer to Figure 2.

### 5.1. Implementing Surgical Interventions in the Management of Male Infertility

Surgical intervention plays a vital role in managing specific forms of male infertility, especially those stemming from physical abnormalities that can be rectified.

#### 5.1.1. Addressing Abnormal Sperm Production: A Focus on Non-Obstructive Azoospermia

Azoospermia is characterized by the lack of sperm in the seminal fluid after centrifugation at 3000× *g* for 15 min followed by a comprehensive microscopic analysis using phase-contrast optics at a magnification of 200× of the resulting pellet [76]. This condition often necessitates a consultation with a urologist. In males with NOA, comprehensive genetic testing is crucial due to the high prevalence (14–19%) of karyotype abnormalities [77]. The most common genetic cause of NOA is Klinefelter syndrome (usually 47,XXY), a disorder that is often undiagnosed, emphasizing the importance of thorough screening in men with azoospermia. This syndrome results in defects in spermatogenesis and hypogonadism characterized by testicular and endocrine system dysfunction. It may lead to poor muscle development, osteopenia in up to 40% of males with Klinefelter syndrome, development of gynecomastia, and potential cognitive or developmental impairments due to testosterone deficiency and infertility [77,78].Additionally, individuals with Klinefelter syndrome are at an increased risk of congenital abnormalities, including metabolic issues, cardiovascular diseases (CVDs) such as venous thromboembolism (VTE), renal problems, breast cancer, and primary extragonadal germ cell tumors [79,80,81,82].

In cases of NOA, treatment options include microdissection testicular sperm extraction (mTESE) and/or surgical repair of palpable varicoceles. Alternatively, using donor sperm and intrauterine insemination (IUI) is generally a more cost-effective approach. However, it is crucial to consider the sociocultural and religious backgrounds of patients when discussing the option of using donor sperm. Acceptance of this option, as well as adoption, varies significantly across different societies. While some cultures and religions encourage adoption as a viable alternative, others strictly forbid pregnancies resulting from donor sperm [83].

For non-obstructive azoospermia, mTESE is the preferred method. It is associated with higher sperm recovery rates (from 16.7 to 45% to 70.8%) and larger sperm counts, with 70 times less testicular tissue removed [84,85,86]. Microsurgical procedures like mTESE are highly complex and should be performed only by an experienced, qualified urologist. A meta-analysis of 117 studies found no difference in the sperm retrieval rate (SRR) between mTESE and standard TESE [87]. SRR is positively correlated with a urologist’s case volume. There is a need for well-designed, adequately powered, randomized controlled trials to demonstrate the advantages of mTESE over standard TESE. Additionally, many patients with NOA should receive hormonal therapy and undergo careful monitoring of their gonadotropin levels [88].

To better illustrate the intricacies of the mTESE procedure, the following Figure 3 provides a detailed view of the surgical process, emphasizing the identification of dilated tubuli, which are often targeted during the extraction.

#### 5.1.2. The Interplay of Varicocele and Assisted Reproductive Technologies

Varicocele, a common cause of male infertility, is a condition characterized by dilation of the veins in the pampiniform plexus, leading to blood stagnation. This can negatively impact testicular function, resulting in reduced sperm quality and quantity. While varicocelectomy, a surgical procedure performed by urologists, can rectify varicoceles and potentially improve fertility outcomes, it is essential to approach this intervention with caution, especially in the context of NOA.

Although some studies suggest that this procedure can enhance the changes in natural conception and reduce the dependence on advanced reproductive technologies such as IVF/ICSI [89,90], the available evidence is of limited quality. Therefore, it is imperative for medical professionals to engage in a comprehensive discussion with patients suffering from NOA and a clinically significant varicocele. This dialogue should encompass the potential benefits and drawbacks of pursuing varicocele repair, ensuring that patients are well informed before proceeding with any treatment intervention [91,92].

Approximately 40% of men with infertility are diagnosed with varicocele. A urologist is able to screen for this condition, make a diagnosis, and assess its severity. Armed with this information, patients can make informed decisions about pursuing surgical correction. The potential benefits can be significant, particularly for those with severe or symptomatic varicoceles. Given its impact on sperm quality and function, reproductive hormone levels, and pregnancy outcomes, varicocele stands as a primary cause of male infertility [93].

According to a study comparing 118 infertile men with 76 healthy controls, the group with varicocele had higher levels of reactive oxygen species and lower total antioxidant capacity [94]. Several factors, including increased venous wall pressure, scrotal heat stress, hypoxia, and renal and adrenal metabolite reflux, likely contribute to these elevated reactive oxygen species levels [95].

Varicoceles also compromise sperm DNA integrity. A study found that the rate of DNA breaks in sperm significantly reduced after surgical correction of a varicocele [96]. Guidelines from the American Urological Association and the American Society for Reproductive Medicine recommend treatment for men with visible varicocele(s), infertility, and abnormal sperm parameters [17].

In the context of varicocele repair in infertile males with oligospermia prior to ART, the increasing use of ARTs has sparked debate about the necessity for varicocelectomy. However, a meta-analysis that encompassed randomized controlled trials and observational studies involving males with clinical varicoceles suggests a different perspective [92,97,98,99,100]. The analysis revealed that varicocelectomy could significantly improve semen parameters in men experiencing deteriorated semen factors [92,97,98,99,100]. This highlights the importance of considering varicocele repair as part of the treatment pathway for these patients [101].

In the case of varicocele repair in men with NOA before ART, the situation is more complex. However, combining varicocelectomy with TESE followed by IVF/ICSI may offer promising outcomes. In particular, varicocelectomy has been found to significantly improve sperm production in men with NOA, leading in some cases to the reappearance of sperm [89].

In terms of improving ART outcomes, varicocelectomy in non-azoospermic infertile men with varicoceles may provide considerable benefits. Not only can it enhance clinical pregnancy rates, live birth numbers, and IVF/ICSI outcomes, but it also represents a less costly treatment option compared to IVF. Current data also suggest potential benefits for men with NOA. However, to fully evaluate the effectiveness of this method, more prospective randomized controlled trials are necessary [102].

In relation to increasing testosterone levels, surgical correction for clinical varicoceles has shown promising results. A study involving 814 men revealed a substantial increase in mean blood testosterone levels—specifically, an increase of 97.48 ng/dL—following the procedure [103]. This finding underscores the importance of counseling hypogonadal men about the potential benefits of surgical correction. Supporting this, a recent meta-analysis reported a modest but statistically significant improvement in average total post-surgical testosterone levels (an increase of 34.3 ng/dL) in hypogonadal men compared to both eugonadal men and untreated controls [104].

#### 5.1.3. Addressing Post-Testicular Obstructions: Surgical and ART Approaches

Obstructive azoospermia (OA) is a condition where no sperm is present in the semen due to an obstruction in the genital tract [105]. This condition is found in 20–40% of men with azoospermia, making it less common than NOA [106,107]. Men with OA typically have normal FSH levels, normal testicular size, and often an enlarged epididymis. The most common site of obstruction in men with primary infertility is the epididymis [108]. Certain types of male infertility, including OA, can be linked to blockages in the reproductive tract. Cases of OA and congenital bilateral absence of the vas deferens necessitate additional genetic testing, as these individuals might have cystic fibrosis or be carriers of the condition [109]. Urologists also need to consider less common but treatable causes of azoospermia such as ejaculatory duct obstruction and retrograde ejaculation, conditions that might require surgical repair of ejaculatory ducts or post-ejaculatory urine testing [106].

These obstructions can arise due to a prior vasectomy or a congenital absence of the vas deferens. In these situations, urologists are equipped to perform surgical interventions aimed at restoring sperm flow. These interventions may encompass microsurgical reconstructive procedures like vasovasostomy (VV) and vasoepididymostomy (VE), which can prove highly beneficial for individuals diagnosed with OA [82]. When these surgical procedures are paired with IVF or ICSI, clinical pregnancy rates can be as high as 65%, with sperm recovery rates potentially reaching 100% [106,110].

A vasectomy reversal can restore the natural flow of sperm, offering couples the potential for natural conception. For men with OA due to congenital bilateral absence of the vas deferens (CBAVD) or other conditions that are not amendable to microsurgical reconstruction, a range of alternative procedures exist. These include percutaneous, open, and microsurgical techniques for sperm retrieval. Procedures such as percutaneous epididymal sperm aspiration (PESA) or TESE may be recommended. Typically, these procedures are used in tandem with ART, including ICSI [111].

Vasectomy reversal is often more cost-effective than IVF or ICSI for achieving parenthood. However, the decision to proceed with a vasectomy reversal should consider the specific circumstances of the couple, including the age of the female partner. If the female partner is between 34 and 40 years old, additional testing may be beneficial due to the window of opportunity for spontaneous conception and IVF/ICSI success [112].

Patients with OA often require a procedure for sperm extraction following the recommended vasal reconstruction. Even though health professionals advise patients with partners over 35 to consider ART, males who have had a vasectomy may find vasal reconstruction more financially viable than undergoing sperm extraction and ART. According to the IVF data from ART’s 2015 annual national summary report, Kapadia et al. noted that IVF cycles resulted in comparable pregnancy and live birth rates among women in matched age groups (35–37, 38–40, and >40 years) [112].

For a more detailed understanding, refer to Table 1, which provides a comprehensive overview of various conditions related to male infertility, their descriptions, the assessments or tests used for their diagnosis, and the treatment or management strategies employed.

A long obstructive interval (over 10–20 years) has often been considered a poor prognostic factor for the success of vasectomy reversal, but recent studies have shown high patency and similar conception rates. When counseling a couple about their treatment options, the financial implications of treatment and insurance coverage should be considered, as vasectomy reversal is often more cost-effective than IVF/ICSI in the long term [113,114].

A reproductive urologist can educate patients about the pros and cons of all treatment options. Reproductive urologists can tailor the treatment of these co-occurring disorders to the needs of each patient.

### 5.2. Pharmacological Approaches in the Management of Male Infertility

Urologists play a crucial role in managing hormonal imbalances that impact fertility, such as hypogonadotropic hypogonadism. Through hormonal therapies, they can restore the body’s normal hormonal balance and improve fertility parameters.

#### 5.2.1. Genital Tract Infections

Infections of the genitourinary system, classified by the World Health Organization (WHO) as male accessory gland infections (MAGIs), can also lead to treatable male infertility. These include urethritis, prostatitis, orchitis, and epididymitis [105]. However, the impact of symptomatic and asymptomatic infections on sperm quality is still a subject of ongoing research [115].

Sexually transmitted infections (STIs) such as Chlamydia trachomatis, Neisseria gonorrhoeae, genital mycoplasma, Trichomonas vaginalis, and various viruses have been linked to male infertility [116]. For instance, men with unexplained infertility are more likely to have HPV than the general population [117,118]. However, the clinical impact of HPV infection in sperm on fertility is yet to be definitively established [119,120,121].

Chronic prostatitis has been shown to negatively affect sperm density, motility, and morphology [122]. Infections with Ureoplasma spp. and Chlamydia trachomatis can reduce sperm count, motility, and morphology and increase DNA damage [123]. Epididymitis, often caused by Chlamydia trachomatis or Neisseria gonorrhoeae in sexually active men under 35 years of age, is also a common cause of male infertility [124].

Treatment strategies for these infections typically involve antibiotics to eliminate or significantly reduce the level of harmful microorganisms in the ejaculate. Other therapeutic approaches include the use of non-steroidal anti-inflammatory drugs (NSAIDs) to manage persistent inflammation, antioxidants, low doses of corticosteroids, mast cell blockers, and other immunomodulatory drugs [125].

#### 5.2.2. Hypogonadotropic Hypogonadism

Primary hypogonadism in men is caused by a problem with the testicles, whereas secondary hypogonadism is caused by issues with the pituitary or hypothalamus. Classifying hypogonadism in this way helps doctors choose the most effective course of therapy. Treatment for secondary hypogonadism may potentially restore fertility and normalize testosterone levels, but in primary hypogonadism, only testosterone therapy is an option, and this medication reduces fertility since it suppresses the hypothalamic–pituitary–gonadal (HGP) axis [126,127].

Grossmann and Matsumoto proposed a new category of adult male hypogonadism, distinguishing between organic and functional hypogonadism [128]. Accordingly, any condition in which the HPG axis has been shown to be disrupted is considered organic hypogonadism requiring standard treatment. However, functional hypogonadism is diagnosed when there is no evidence of organic changes in the HPG axis, so all concomitant diseases must be clarified first [128].

In the era of IVF and ICSI, it is critical that urologists help maximize sperm production in subfertile men. Gonadotropin-releasing hormone (GnRH), LH, and FSH are three main treatments specifically approved for infertility by regulatory authorities [129,130]. A Cochrane Review revealed that the live pregnancy rate was significantly higher in treated men compared to the placebo or untreated group. The spontaneous pregnancy rate was also found to be higher in treated men [131].

There is a growing body of data suggesting that men often have worse outcomes than women after infection with severe acute respiratory syndrome coronavirus 2 (SARS-CoV-2) (COVID-19), although the prevalence is the same in all sexes [132]. Androgens may play a role in explaining these sex differences, as testosterone has been shown to regulate tissue expression of angiotensin-converting enzyme 2 (ACE2) and transcription of a transmembrane protease, serine 2 (TMPRSS2), both of which are involved in the process of cellular internalization of the virus [133,134]. Not surprisingly, SARS-CoV-2 may contribute to decreased testosterone and sperm production, as ACE2 is expressed in numerous tissues, including the testis. There is evidence that the virus can cause a localized, intense inflammatory response in the testis, which can then promote the development of viral orchitis and lead to vasculitis or an autoimmune response, both of which can damage the testis and impair testosterone production [135].

There is still much debate about whether or not low testosterone directly contributes to the poorer results with COVID-19. It cannot be ruled out that low testosterone in the acute phase of viral infection may represent an adaptive and resilient mechanism to mitigate external impairments by eliminating testosterone-dependent functions such as reproduction and/or physical and sexual activity that are not needed when the physical state deteriorates [136,137,138,139,140]. Men who have recovered from SARS-CoV2 infection should be carefully monitored to rule out the possibility of long-term andrological effects such as decreased sperm count or testosterone levels [135,141].

#### 5.2.3. Endocrine Disorders

Endocrine disorders, including thyroid dysfunction, hyperprolactinemia, diabetes, and insulin resistance, can negatively impact male fertility. These conditions can cause alterations in spermatogenesis, lower sperm quality, and erectile dysfunction [142]. For instance, dysfunction of the thyroid can result in decreased levels of the sex hormone-binding globulin (SHBG), which may in turn lead to secondary hypogonadism [143,144]. Hyperprolactinemia can suppress the release of gonadotropin-releasing hormone, reducing the synthesis of FSH and LH and directly inhibiting spermatogenesis and steroidogenesis [145]. Diabetes and insulin resistance, especially when co-occurring with obesity and metabolic syndrome, can affect spermatogenesis and sperm DNA integrity [146,147].

Furthermore, the role of gonadotropin therapy in relation to pregnancy rates has been explored in a few studies. These analyses primarily focused on randomized, controlled trials that utilized clinically relevant outcomes such as pregnancy rates rather than solely on semen parameters. While the limited number of studies available hindered the attainment of significant statistical power for conclusive findings, the combined data analysis revealed a significant increase in pregnancy rates within a 3-month period post-gonadotropin therapy: 13.4% compared to 4.4% (with an odds ratio (OR) of 3.03 and a 95% confidence interval (CI) ranging from 1.30 to 7.09). Of these, three trials specifically assessed spontaneous pregnancy rates, showcasing a rate of 9.3% in the treatment group versus 1.7% in the control group (with an OR of 4.18 and a 95% CI ranging from 1.38 to 12.37) [148]. Despite the known links between these endocrine problems and male infertility, their impact on ICSI outcomes or the potential benefits of treatment before ICSI remain unexplored.

#### 5.2.4. Addressing Erectile Dysfunction and Premature Ejaculation for Fertility Enhancement

Urologists play a crucial role in managing conditions such as erectile dysfunction and premature ejaculation, which can significantly impact a man’s fertility. Erectile dysfunction, characterized by the inability to achieve or maintain an erection suitable for intercourse, can prevent natural conception. Similarly, premature ejaculation, where ejaculation occurs sooner than desired during sexual activity, can also hinder successful conception.

Urologists can prescribe a range of medications to address these issues. For erectile dysfunction, treatments may include phosphodiesterase type 5 inhibitors (PDE5is) like sildenafil and vardenafil, which enhance the effects of nitric oxide, a natural chemical the body produces to relax penis muscles and increase blood flow [149,150].

Sofikitis et al. conducted a study exploring the potential of PDE5is as an adjunctive therapy for male infertility [149]. Their research revealed that PDE5is had a positive impact on the secretory function of Leydig and Sertoli cells and were linked to the regulation of contractility in the tunica albuginea of the testes and epididymis. Moreover, PDE5is have been shown to enhance sperm motility by augmenting prostatic secretory activity and to play a pivotal role in the regulation of sperm capacitation [150].

A recent meta-analysis that encompassed nine RCTs involving 1211 participants from various nations delved deeper into the effects of PDE5is on male fertility [151]. Participants were administered varying dosages of PDE5is and underwent evaluations of their sperm parameters both before and during the therapy. The goal was to compare the outcomes of males treated with PDE5is to those given a placebo. The findings indicated a significant improvement in sperm concentration, motility, and morphology. However, two limitations were noted: the smaller size of prior RCTs and the omission of individual patient demographic data, which could introduce bias [151].

For premature ejaculation, selective serotonin reuptake inhibitors (SSRIs) such as dapoxetine or local anesthetic creams and sprays can be used to delay ejaculation [152,153]. By effectively managing these conditions, urologists can significantly improve a couple’s chances of achieving successful natural conception, reducing the need for more invasive fertility treatments.

### 5.3. Role of Urologists in Managing the Impact of Lifestyle Factors on Male Fertility and Overall Health

Lifestyle factors significantly impact male fertility, and thus urologists often recommend certain lifestyle modifications to enhance fertility outcomes. These include regular exercise, a healthy diet, maintaining optimal weight, avoiding exposure to environmental toxins, and limiting intake of tobacco and alcohol [154,155]. Obesity has been suggested to negatively impact sperm quality, possibly through mechanisms involving hormonal imbalances, increased reactive oxygen species (ROS) levels, and elevated testicular temperatures [146,156]. While some studies have suggested links between obesity and reduced fertility outcomes, the findings are inconsistent. Nevertheless, weight loss and regular physical activity have been associated with improved sperm parameters and hormonal profiles in some studies [157,158].

About a third of men of reproductive age are smokers, a habit that has shown a consistent negative correlation with sperm parameters [159]. Smoking cessation can reportedly improve sperm quality, making anti-smoking initiatives beneficial [160]. Moderate alcohol consumption does not adversely affect sperm parameters [161]. However, excessive intake (defined as >2 drinks per day [162]) may harm male fertility, possibly due to a decline in testosterone levels, which can be reversed by abstaining from alcohol [163]. Exposure to environmental and occupational toxicants, including endocrine-disrupting chemicals (EDCs), can impair spermatogenesis [164]. EDCs disrupt hormonal balance, which can adversely impact sperm production and quality [165]. Prolonged exposure to extreme environmental conditions like high temperatures and intense noise may also negatively affect fertility [166].

There is a growing recognition of a direct link between men’s reproductive and overall health. Genetic issues contribute to infertility in about 15% of males [167]. Infertile men have higher incidences of certain cancers and chronic diseases like diabetes and coronary heart disease [168,169]. Additionally, abnormalities in semen parameters have been correlated with an increased risk of various health conditions [170]. Infertility in males has also been associated with increased mortality, highlighting the importance of referring infertile men to a urologist for early identification and management of potential health risks [170]. Through their surgical expertise, urologists can address many causes of male infertility and subfertility (Table 1).

### 5.4. The Reproductive Urologist’s Role in Special Circumstances

#### 5.4.1. Addressing Idiopathic Recurrent Pregnancy Loss

Recurrent pregnancy loss (RPL) affects 2–5% of couples, with half of the cases having an unknown cause [171,172]. Research suggests that male factors, such as sperm DNA integrity, may contribute to RPL. High levels of DNA fragmentation in sperm may be irreversible and hinder embryonic development [172,173]. This finding can guide treatment, with lifestyle changes and antioxidant treatment recommended to lower DNA fragmentation levels [174].

Additionally, chromosomal abnormalities or sperm aneuploidy in males can be associated with RPL. Even with normal sperm analysis results, these abnormalities may be present. Therefore, men whose partners have experienced multiple miscarriages should undergo a sperm FISH test [175]. The results can guide clinical decisions and counseling. For couples with detected sperm aneuploidy, preimplantation genetic diagnosis for monogenic/single gene abnormalities (PGT-M) with IVF is an option. If RPL occurs with no female factors, the male factor (e.g., DNA fragmentation) should be considered. Various tests can help determine the presence of sperm aneuploidy or DNA fragmentation, influencing future treatment decisions (e.g., IUI vs. IVF vs. ICSI) [30,176].

#### 5.4.2. Overcoming Recurrent Failures in Assisted Reproductive Technologies

Accumulating evidence has underscored the role of oxidative stress and sperm DNA fragmentation (SDF) in the pathophysiology of male infertility [177]. Elevated SDF levels can contribute to IUI failure [178]. High SDF levels, as indicated by a DNA fragmentation index (DFI) >30% in the sperm chromatin structure assay (SCSA), are associated with lower pregnancy and delivery rates [179]. Other SDF tests, such as terminal deoxynucleotidyl transferase-dUTP nick-end labeling (TUNEL), also predict poor IUI outcomes, suggesting that couples with high SDF levels should consider IVF instead [180,181]. However, not all SDF tests, such as sperm chromatin dispersion (SCD), correlate with IUI outcomes [182]. Therefore, consultation with a reproductive urologist can help select the appropriate SDF test and decide whether to proceed with IVF.

Chromosomal aneuploidy and structural abnormalities can also cause IVF failure. Even men with normal sperm motility and density may have abnormal FISH findings, indicating the need for aneuploid sperm testing [183]. High SDF levels can also lead to worse implantation rates and embryonic development, although pregnancy rates may remain the same [184]. However, selecting testicular sperm for ICSI can improve live birth rates in men with high SDF levels [185]. Therefore, after IVF failure, sperm aneuploidy testing and SDF testing should be discussed with a male infertility specialist to guide the next steps in treatment.

#### 5.4.3. Implications of DNA Fragmentation on ART Outcomes: Strategies for Sperm Isolation and Recovery

Sperm DNA integrity has emerged as a significant factor in both spontaneous and assisted conception, with SDF assays serving as a complementary diagnostic tool to standard semen analysis [186,187]. High SDF levels are associated with decreased conception rates in both IVF and ICSI [188,189]. However, live birth rates are higher with ICSI than traditional IVF in couples with high SDF [190,191].

High SDF levels also increase the risk of pregnancy loss after IVF and ICSI [192,193,194]. Therefore, ICSI is recommended over traditional IVF in couples with high SDF despite the associated risk of miscarriage [195]. Lifestyle changes and counseling can help manage high SDF levels, and surgical procedures such as varicocelectomy or sperm retrieval may also be beneficial [96,196]. However, the use of testicular or epididymal sperm, which have lower SDF levels than ejaculated sperm, in conventional fertility treatments is still controversial [196,197].

Current guidelines from the European Academy of Andrology [154], the European Association of Urology (EAU) [14], and the American Urological Association/American Society for Reproductive Medicine (AUA/ASRM) [17] recommend the use of surgically collected sperm for ICSI in certain cases. Collaboration between reproductive gynecologists and reproductive urologists is crucial in determining the best treatment approach.

## 6. Uncovering Conditions That Might Affect the Offspring’s Health

Understanding the potential genetic and epigenetic risks associated with certain male infertility conditions is crucial for reproductive urologists. These risks can have profound implications for the health of the offspring. This section delves into some of these conditions and their potential impact on the health of the child.

### 6.1. Advanced Paternal Age

Although little is known about paternal characteristics of births, men nowadays are fathering children at older age than before [198]. The aging process deteriorates male infertility, affecting multiple sperm parameters and changing the profile of reproductive hormones [199]. There are several reports relating the advanced paternal age with genetic abnormalities, such as DNA mutations and chromosomal aneuploidies, and epigenetic alterations [200]. Fertility and reproductive outcomes, such as the success rate of IVF and ICSI, and premature birth rates are linked with the father’s age [201]. The paternal advanced age is connected with multiple offspring diseases, including bipolar disorders, schizophrenia, autism, and pediatric leukemia [202,203,204]. Urologists can inform the infertile male of the distressing associations between advanced paternal age and an increase in their offspring’s diseases so that they can be efficiently conducted during their reproductive age [205]. 

### 6.2. Effects of Klinefelter Syndrome on the Health of the Offspring

In Klinefelter syndrome, the testis exhibits a mosaic situation in which the majority of tubules have 46,XXY spermatogonia, whereas spermatogonia with a normal chromosome value (46,XY) are present in a few tubules [206]. This suggests that spermatozoa from KS patients probably originated from euploid spermatogonia. Accordingly, the data found in scientific research do not suggest an increased likelihood of fathering a child with KS compared to infertile men who have a normal karyotype [206]. Several cases of 47,XXY fetuses/newborns have been documented, but a total of more than 200 healthy children have been born to KS fathers worldwide [207,208,209]. Preimplantation genetic diagnosis (PGD) or prenatal genetic testing remains controversial [210], although there is promising evidence that KS children do not appear to be affected by the father’s genetic disease.

### 6.3. Consequences of Y-Chromosome Microdeletions

Genetic counseling is required because Y deletions are passed to sons after conception. In most cases, the microdeletion is identical in father and son [211], but in rare cases the son may have a larger deletion [212]. Because of the different genetic background of the son and the presence or absence of environmental elements that may adversely affect reproductive function, it is impossible to predict the full extent of spermatogenic failure (still in the azoo/oligo zoospermia range). Regarding sex chromosomes, a substantial proportion of spermatozoa from males with complete AZFc deletion are nullisomic [213,214], indicating a potential risk for offspring to develop 45,X0 Turner syndrome and other phenotypic abnormalities associated with sex chromosome mosaicism, including ambiguous genitalia [215]. Despite this potential risk, phenotypically normal children are born to fathers who have Yq microdeletions [211,215]. This may be due to the fact that embryos with a 45,X0 karyotype have a lower implantation rate and thus a higher risk of spontaneous abortion.

### 6.4. Impact of ICSI on the Offspring’s Health

ICSI has outpaced all other forms of fertilization in recent years, and its use has increased most rapidly in cycles where male infertility was not a problem [190]. However, some research suggests that ICSI poses a greater risk to the developing embryo than other reproductive methods. The use of ICSI remained significantly associated with an increased risk of birth anomalies in a substantial observational study of more than 300,000 births in Australia [216]. In a multicenter European cohort study of approximately 1500 infants, Bonduelle et al. demonstrated that the only offspring at higher risk for significant congenital anomalies were those conceived via ICSI [217]. The potential negative impact of this alarming increase in birth anomalies on future generations is debatable. In individuals with infertility not due to sperm deficiency, ICSI has been associated with lower birth weight [218] and increased autism risk [219]. Although the American Society for Reproductive Medicine has found an association between ICSI and a slightly increased risk of poor outcomes in children, the relationship between this risk in patients with infertility without male factors remains unclear [220]. Reproductive urologists should work with female infertility specialists to maximize male fertility through less invasive reproductive treatments that may result in healthier offspring.

The health of the offspring is a paramount concern for couples seeking fertility treatments. Understanding the potential risks associated with male infertility conditions and the treatments employed is essential. Reproductive urologists play a pivotal role in counseling and guiding couples through these complexities, ensuring the best possible outcomes for both the parents and the offspring.

## 7. Barriers to Urologist Involvement in the Era of Assisted Reproductive Technology

The advent of ART has revolutionized the field of reproductive medicine, offering hope to countless couples struggling with infertility. However, the role of urologists, especially in the context of male infertility, remains underemphasized and often overlooked. Several barriers hinder the full involvement of urologists in this evolving landscape:Limited availability and distribution: One of the primary challenges is the limited number and distribution of specialized male reproductive urologists. Certain regions, especially rural or underserved areas, lack access to these experts, making it challenging for patients to seek specialized care [16].Lack of awareness and education: There is a significant knowledge gap among healthcare professionals and the general public regarding the role of urologists in male infertility. This lack of awareness leads to under-referral and underutilization of urologist services, with many cases of male infertility going undiagnosed or mismanaged [16].Financial constraints: The high costs associated with infertility treatments coupled with the absence of comprehensive health insurance coverage for such services in many regions deter many from seeking specialized care. This financial barrier is further exacerbated by the often-prohibitive costs of ART procedures [221].Misconceptions about infertility: A prevailing misconception exists that infertility is primarily a female-centric issue. This skewed perspective often results in a disproportionate focus on female infertility, sidelining male factors that, in reality, contribute to almost half of all infertility cases [16].Lifestyle and environmental factors: The modern era has seen a decline in male fertility attributed to various lifestyle and environmental factors. However, the role of urologists in addressing these factors and guiding patients toward healthier lifestyles remains underemphasized [12].Rapid advancements in ART: The swift advancements in ART techniques have sometimes overshadowed traditional diagnostic and therapeutic approaches. As a result, the emphasis on understanding and treating the root causes of male infertility, where urologists play a pivotal role, has been diminished [222].

In light of these barriers, it becomes imperative to re-evaluate and emphasize the role of urologists in the era of ART. Their expertise in diagnosing, managing, and treating male infertility is invaluable. Addressing these challenges requires a multifaceted approach encompassing awareness campaigns, educational initiatives, policy changes, and collaborative efforts between various stakeholders in the field of reproductive medicine. Only by recognizing and addressing these barriers can we truly harness the potential of urologists in optimizing fertility outcomes in the era of ART.

## 8. The Future of Urologists: What Place Do They Have in the Era of ART?

The current standard treatment for infertility places a significant burden on women. The male examination is quick, easy, and non-invasive compared to the female study. The identification and treatment of male causes of infertility means that the ART technique can be downgraded from ICSI to IUI or even eliminated. Decreased ovarian reserve in women is a problem. Declining sperm count is another problem. According to a meta-analysis, sperm counts fell by 50% between 1973 and 2011 [223]. Many women are postponing the desire to have children for social, economic, and cultural reasons, so more and more women need ART at a younger age than in the past. The fact that urologists assess and treat the male infertility factor will make life easier for ART clinics, as fewer women will seek their services.

## 9. Conclusions

When a couple first seeks help for infertility, they usually turn to an obstetrician/gynecologist for diagnosis and treatment. In some instances, sending the patient to a reproductive urologist sooner rather than later may increase the chances of a successful pregnancy. The contribution of the urologist is crucial. Before ART therapy, all men should be evaluated for fertility to rule out more serious underlying pathologic disorders. Once the decision is to use ART, the urologist and obstetrician/gynecologist should work together to choose the safest, most cost-effective, and most appropriate method. Semen analysis abnormalities, failed IVF, failed IUI, palpable varicocele, or idiopathic RPL are all reasons to see a reproductive urologist. The urologist will administer medications to improve sperm characteristics if needed. They will also perform the appropriate surgery in cases suitable for surgical intervention. Recent research efforts have demonstrated very vividly that the mission of the sperm is not simply the fertilization of the oocyte. Sperm bring genetic (i.e., DNA content) and epigenetic factors (i.e., reproducing elements of the centrosome and the protein matrix) that have a major impact on early embryonic development and capacity for implantation. The urologist is the appropriately trained physician in testicular and sperm physiology to provide their knowledge to the infertile couple.

## Figures and Tables

**Figure 1 medicina-59-01835-f001:**
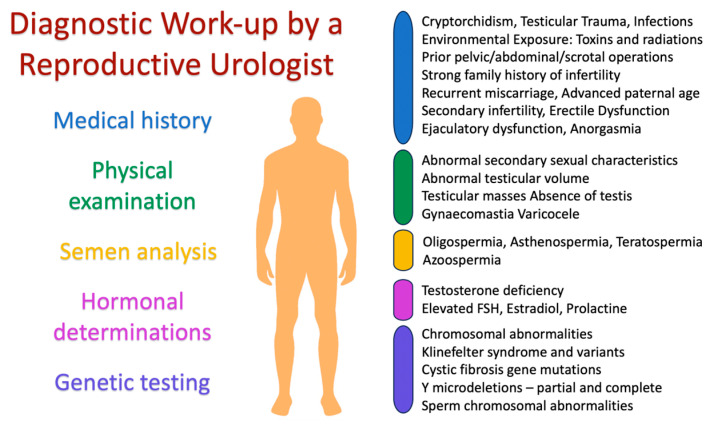
Diagnostic work-up by a reproductive urologist.

**Figure 2 medicina-59-01835-f002:**
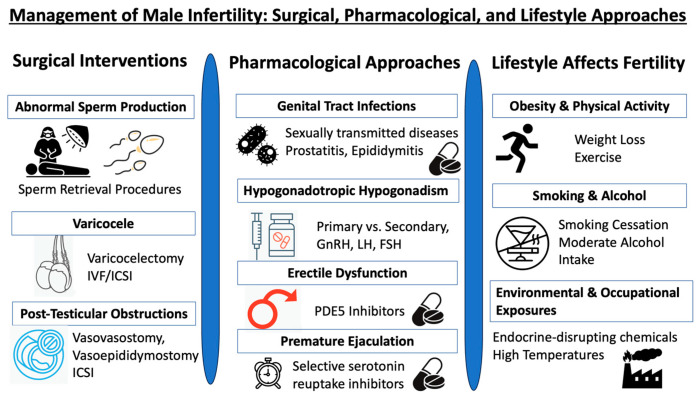
Triad of male infertility management: surgical, pharmacological, and lifestyle approaches.

**Figure 3 medicina-59-01835-f003:**
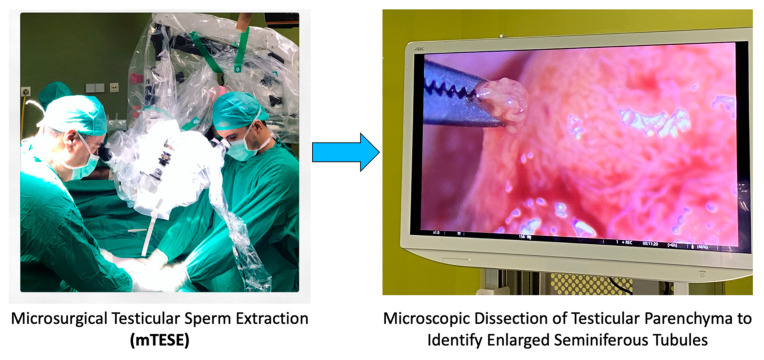
Surgical procedure of mTESE highlighting dilated tubuli.

**Table 1 medicina-59-01835-t001:** Comprehensive overview of male infertility conditions, diagnostics, and management strategies.

Categories/Condition	Description	Assessment/Tests	Treatment/Management
Male infertility evaluation	Evaluation of male infertility	Semen analysis and genetic testing	Individualized based on the findings
Semen analysis abnormalities	Any irregularities found during semen analysis such as low sperm count, poor motility, or morphology	Semen analysis	A reproductive urologist can diagnose the underlying cause of the abnormalities and suggest appropriate treatments
Azoospermia	Complete absence of sperm in the semen	Semen analysis	A reproductive urologist can help identify and treat the cause
Leukocytospermia	Presence of white blood cells in the semen, which may indicate an infection or inflammation	Semen analysis	A reproductive urologist can determine the cause of the inflammation or infection and prescribe appropriate treatments
Idiopathic recurrent pregnancy loss	Pregnancy loss due to unknown male factors such as sperm DNA integrity	SDF tests like TUNEL, SCSA, FISH analysis, and the sperm aneuploidy test	Lifestyle changes, antioxidant treatment, and PGT-M with IVF
Recurrent ART failure	Repeated failure of ART procedures such as IVF	SDF tests like TUNEL and SCSA	A reproductive urologist can evaluate potential male factor causes for the failure of ART procedures and suggest alternatives or adjustments to the treatment plan
ART results and DNA fragmentation	Impact of sperm DNA fragmentation on conception rates in IVF and ICSI	SDF assays	Lifestyle changes, varicocelectomy, and sperm retrieval
Klinefelter syndrome	Infertility caused by additional X chromosome in males	Karyotyping	Assisted reproductive techniques and genetic counseling
Y-chromosome microdeletions	Infertility due to deletions on the Y chromosome	Y deletion test	Assisted reproductive techniques, genetic counseling, and potential ICSI
Advanced paternal age	Infertility associated with advanced paternal age	General health check-up and genetic tests	Counseling and possibly assisted reproductive techniques

Abbreviations: ART, assisted reproductive technology; FISH, fluorescence in situ hybridization; PGT-M, preimplantation genetic testing for monogenic/single-gene defects; IVF, in vitro fertilization; ICSI, intracytoplasmic sperm injection; SDF, sperm DNA fragmentation; TUNEL, terminal deoxynucleotidyl transferase dUTP nick end labeling; SCSA, sperm chromatin structure assay.

## Data Availability

Not applicable.

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
