# Peer review of "From Diagnosis to Treatment: Comprehensive Care by Reproductive Urologists in Assisted Reproductive Technology"

_medicina, 2023, doi:10.3390/medicina59101835_

Round 1

Reviewer 1 Report

the authors did an excellent job discussing male fertility as a possible cause of couple infertility.

it is not clear if the authors intended to create a narrative review or a book chapter

matherial and methods? type of study? in which databases were performed the research? how long time ? inclusion exclusion criteria for the articles? 

For chapter 3, perhaps a graphic representation of the role of the urologist in the management of male infertility would be useful

a comparison would be useful (ICSI/IFV) as a  table for the various male conditions, with the year of the study, the number of participants, with p/OR

maybe some intraoperatory photos regarding the authors experience in this field ( varicocel reparation/ vasovasostomy (VV) or vasoepididymostomy (VE)

good luck!

Author Response

Responses to Reviewer 1

Reviewer Comment:

The authors did an excellent job discussing male fertility as a possible cause of couple infertility.

Response:

Thank you for your positive feedback. We aimed to provide a comprehensive overview of male fertility and its implications in couple infertility, and we are pleased to hear that our efforts resonated with you.

Reviewer Comment:

It is not clear if the authors intended to create a narrative review or a book chapter.

Response:

We apologize for any confusion. Our intention was to create a comprehensive narrative review on the topic. We understand the depth and breadth of the content might resemble a book chapter, and we will ensure to clarify the format in our introduction and methodology sections.

Reviewer Comment:

Material and methods? Type of study? In which databases were performed the research? How long time? Inclusion exclusion criteria for the articles?

Response:

We appreciate your feedback on this aspect. We realize that we did not provide a clear "Materials and Methods" section, which is essential for a narrative review. In our revised manuscript, we will include a detailed section outlining the type of study, databases searched, duration of the search, and the inclusion and exclusion criteria for the articles reviewed.

Reviewer Comment:

For chapter 3, perhaps a graphic representation of the role of the urologist in the management of male infertility would be useful.

Response:

Thank you for the suggestion. We agree that a visual representation can enhance the understanding of the urologist's role in managing male infertility. We will incorporate a graphic representation in Chapter 3 to elucidate this aspect further.

Reviewer Comment:

Maybe some intraoperative photos regarding the authors' experience in this field (varicocele reparation/ vasovasostomy (VV) or vasoepididymostomy (VE)).

Response:

We appreciate the suggestion. We believe that including intraoperative photos can provide valuable insights into the surgical procedures. We have added a figure from our experiences with mTESE to enhance the manuscript's visual appeal and provide a practical perspective.

Reviewer Comment:

Good luck!

Response:

Thank you for your constructive feedback and well wishes. We value your insights and have made efforts to address all the points raised. We believe these changes will enhance the quality and clarity of our manuscript.

Reviewer 2 Report

The manuscript “From Diagnosis to Treatment: The Comprehensive Care of Reproductive Urologist in Assisted Reproductive Technology” is aims to optimize the involvement of urologists in the management of male infertility and improve outcomes for couples seeking fertility treatment.

Unfortunately, the study has serious limitans that I exposed below.
It is not clear what novel information this study provides. As was cited by the authors, “the role of urologists in the era of assisted reproductive technology is vital for diagnosing and managing male infertility”.

Author Response

Responses to Reviewer 2

Reviewer Comment:

The manuscript “From Diagnosis to Treatment: The Comprehensive Care of Reproductive Urologist in Assisted Reproductive Technology” aims to optimize the involvement of urologists in the management of male infertility and improve outcomes for couples seeking fertility treatment.

Response:

Thank you for summarizing the aim of our manuscript. We believe that a comprehensive understanding of the urologist's role in the context of Assisted Reproductive Technology (ART) is crucial for both medical professionals and patients.

Reviewer Comment:

Unfortunately, the study has serious limitations that I exposed below.

Response:

We appreciate your feedback and are committed to addressing the concerns you've raised. We aim to improve the manuscript based on your valuable insights.

Reviewer Comment:

It is not clear what novel information this study provides. As was cited by the authors, “the role of urologists in the era of assisted reproductive technology is vital for diagnosing and managing male infertility”.

Response:

Thank you for pointing this out. While we acknowledge that the importance of urologists in the realm of ART is well-recognized, our manuscript delves deeper into the nuances of their involvement. We aim to provide a comprehensive overview of the various strategies, interventions, and innovations that reproductive urologists bring to the table. This includes not just the diagnostic aspects but also the therapeutic and collaborative roles they play in conjunction with other fertility specialists. Our manuscript also emphasizes the need for a multidisciplinary approach, highlighting the synergistic benefits of integrating urological expertise into ART protocols. We will work on clarifying these points further in our revised manuscript to emphasize the unique contributions of our study.

Reviewer 3 Report

In order to enhance the emphasis on reproductive urologists in the assisted reproductive process, this manuscript presents a comprehensive review of their role in managing male infertility. It outlines their expertise in diagnosing and managing male infertility, addressing reversible causes, and performing surgical techniques such as sperm retrieval. The study also aims to identify barriers to urologists’ involvement in ART process and suggests potential strategies to overcome these obstacles.

The authors provide a summary and discussion of the role of urologists in various aspects of assisted reproduction, including evaluation and diagnosis, addressing male infertility, enhancing ART outcomes, and uncovering conditions that might affect offspring’s health. They also discuss the important role of urologists in uncovering conditions that might impact offspring’s health. The present review was well written, but there are some details that need to be addressed:

1.    The reasons behind the lack of emphasis on urologists’ role in assisted reproduction has not been discussed sufficiently and needs further exploration.

2.    Although Table 1 provides a detailed summary of the role of urologists in ART, it is suggested to create a schematic diagram summarizing their role throughout the entire ART process. This diagram should include the evaluation of male infertility prior to ART, their involvement during the process, and the assessment on offspring’s health.

3.    In the section “5. Uncovering conditions that might affect offspring’s health,” the last paragraph discusses the impact of ICSI on offspring, but it lacks an introduction and conclusion of section 5. In the subsection on the impact of advanced paternal age on offspring’s health, the authors only cite a review to illustrate the issue. It is recommended to add the latest research literature for reference.

This review was well written. Minor editing of English language is required. 

Author Response

Responses to Reviewer 3

Reviewer Comment:

In order to enhance the emphasis on reproductive urologists in the assisted reproductive process, this manuscript presents a comprehensive review of their role in managing male infertility.

Response:

We appreciate your acknowledgment of the manuscript's intent and the importance of highlighting the role of reproductive urologists in the assisted reproductive process.

Reviewer Comment:

  1. The reasons behind the lack of emphasis on urologists’ role in assisted reproduction has not been discussed sufficiently and needs further exploration.

Response:

Thank you for pointing this out. We recognize the importance of delving deeper into the reasons for the underemphasis on urologists' roles in assisted reproduction. In our revised manuscript, we will expand on this topic, discussing historical, educational, and systemic factors that might have contributed to this oversight and how the landscape is changing in contemporary practice.

Reviewer Comment:

  1. Although Table 1 provides a detailed summary of the role of urologists in ART, it is suggested to create a schematic diagram summarizing their role throughout the entire ART process. This diagram should include the evaluation of male infertility prior to ART, their involvement during the process, and the assessment on offspring’s health.

Response:

This is an excellent suggestion. A visual representation can indeed provide a clearer and more immediate understanding of the urologist's role throughout the ART process. We will work on creating a comprehensive schematic diagram that encapsulates the evaluation, involvement, and postART assessment roles of urologists, ensuring it complements the information provided in Table 1.

Reviewer Comment:

  1. In the section “5. Uncovering conditions that might affect offspring’s health,” the last paragraph discusses the impact of ICSI on offspring, but it lacks an introduction and conclusion of section 5. In the subsection on the impact of advanced paternal age on offspring’s health, the authors only cite a review to illustrate the issue. It is recommended to add the latest research literature for reference.

Response:

We appreciate your detailed feedback on Section 5. We acknowledge the need for a more structured introduction and conclusion to provide context and summarize the key points discussed. We will revise this section to provide a clearer framework. Additionally, we will incorporate the latest research literature on the impact of advanced paternal age on offspring's health to ensure our manuscript is uptodate and comprehensive.

Reviewer 4 Report

I would like to thank the authors for their contribution. However, there is a minor revision to be taken care of before final publication which has been addressed in the comment section for authors.

Comment to authors

Abstract: Well written and gains the attention of readers.

Introduction:

Line 68: No need to mention the sentence “In conclusion”.

Line 107, Line 144: The Section 3 references between [17] to [57] are missing. References sequences should be in order manner. Re-check the reference sequences. It seems wrong citation for reference [75] in this location, re-check, please.

Line 126: The author should elaborate more about the role of Testicular biopsy in male infertility evaluation, It is important to mention its clear indication based on evidence-based medicine because unfortunately, many abuses such interventions even from obe/gyn teams perform such procedures, which is not acceptable ethically and professionally and should be always done by reproductive urologists. Use the following articles to enhance the role of Testicular biopsy in male infertility diagnosis: PMID: 33295257, PMID: 34511305

Line 128: Genetic testing, the author should elaborate more about such tests and the recent guidelines clearly showing their indication. Use the following articles to enhance the role of Genetic testing in male infertility diagnosis: PMID: 33295257, PMID: 34511305

Line 311: It is better to describe accurately the definition of azoospermia, add the statement:  Azoospermia, marked by the absence of sperm in semen after centrifugation at 3,000 g for 15 minutes and a thorough microscopic examination by phase contrast optics at ×200 magnification of the pellet”.

Reference: WHO laboratory manual for the examination and processing of human semen Sixth edition. 2021. https://www.who.int/publications/i/item/9789240030787

Line 316 to 320: It is also important to add the risk of congenital abnormalities including metabolic, cardiovasculardiseases (CVD) including venous thromboembolism (VTE), renal problems, and breast and primary extragonadal germ cell tumors among cases with Klinefelter syndrome. Use the references PMID: 28258556, PMID: 27582022, PMID: 21241366, PMID: 33005196.

Line 322, Lines 336 to 339: The author mentioned the role of varicocele repair in NOA, however, needs further support from the literature. The available evidence is of limited quality, and it is crucial to engage in a comprehensive discussion with patients suffering from NOA and a clinically significant varicocele about the potential advantages and drawbacks of pursuing varicocele repair before proceeding with any treatment intervention. Use the following references to emphasize the role of such intervention in cases of NOA and its clear indication. PMID: 30291690, PMID: 26680033

Line 322 to 323: It is important to discuss the sperm donor based on the sociocultural and religious background as acceptance for such option as well as adoption. Some societies encourage adoption as an alternative and in some societies, donor sperm pregnancies are strictly forbidden. Use reference PMID: 11844368

Table 1: The category/condition (Idiopathic Recurrent Pregnancy Loss), the growing evidence now supporting the use of SDF tests, see reference PMID: 33422457.

Table 1: The category/condition (Y-chromosome Microdeletions), the author must mention that Sperm retrieval from category AZFa and AZFb regions is unsuccessful, while from category AZFc region may associated with a success rate of sperm retrieval.

Line 532:  the role of gonadotropin therapy compared to pregnancy rate was investigated in a few studies, the analysis primarily focused on randomized, controlled trials that utilized clinically relevant outcomes, such as pregnancy rates, as opposed to relying solely on semen parameters. Although the number of studies available was insufficient to achieve significant statistical power for drawing definitive conclusions when examining the combined data, the analysis identified a noteworthy increase in pregnancy rates within a 3-month period following gonadotropin therapy: 13.4% compared to 4.4% (with an odds ratio [OR] of 3.03 and a 95% confidence interval [CI] ranging from 1.30 to 7.09). Among these studies, three trials specifically evaluated spontaneous pregnancy rates, revealing a rate of 9.3% in the treatment group versus 1.7% in the control group (with an OR of 4.18 and a 95% CI ranging from 1.38 to 12.37). Please use this reference PMID: 18423250

Line 535: Please use this reference to discuss the effect of PDE5i on semen parameters.

Limited Effect on Semen Parameters: Research suggests that PDE5 inhibitors, such as sildenafil (Viagra), tadalafil (Cialis), and vardenafil (Levitra), generally have a limited direct effect on semen parameters like sperm count, motility, and morphology. They are not designed to enhance fertility or improve sperm quality. A mong mean age of the study participants was 35.4 years, with an average sperm concentration of 68.7 million sperm per milliliter (±32.4) and an average sperm motility of 50.38% (±8.41). Over time, all three groups (control, sildenafil, tadalafil) exhibited trends of decreased sperm motility, but these changes did not reach statistical significance. Furthermore, there were no notable variations among the three groups in terms of the acrosome reaction following 120 minutes of drug exposure. Importantly, the highest semen concentration observed after oral administration of sildenafil (100 mg) or tadalafil (20 mg) did not have a substantial impact on either sperm motility or the acrosome reaction (PMID: 33943054). Another study reported the result of a prospective, randomized, double-blind, crossover clinical trial, after administering sildenafil, a notable rise in sperm progressive motility (with a median value of 37.0%) was observed when compared to the baseline measurement. In contrast, following tadalafil administration, there was a significant decrease in this parameter (with a median value of 21.5%) as compared to the baseline value (PMID: 17544419).

Line 591 to 599: The growing evidence now supports the use of SDF tests, see reference PMID: 33422457.

Line 608 to 610: Please see updated guidelines regarding Sperm DFI and compress different assays in DFI test, see reference PMID: 33422457.

Author Response

Dear Reviewer 4,

We would like to express our sincere gratitude for your thorough review and constructive feedback on our manuscript titled “From Diagnosis to Treatment: The Comprehensive Care of Reproductive Urologist in Assisted Reproductive Technology.” Your insights have been invaluable in refining our work and ensuring its scientific rigor.

In conclusion, we have made all the changes you recommended and believe that these revisions have significantly improved the quality and depth of our manuscript. We are grateful for the time and effort you invested in reviewing our work. Your expertise has been instrumental in enhancing the quality of our research.

Once again, thank you for your invaluable feedback. We hope that the revised manuscript now meets the standards of the journal and look forward to any further suggestions or recommendations you might have.

Round 2

Reviewer 1 Report

Thank you for your answers. 

The authors really improved the first version. 

Good luck 

Reviewer 3 Report

All suggested changes have been made. The reviewer finds that the current version of the manuscript is acceptable for publication at Medicina. This has a significantly positive impact on Reproductive Urologists' increased involvement in assisted reproductive management, leading to greater benefits for patients.